# Synthesis and Structural Characterization of Phosphanide Gold(III)/Gold(I) Complexes and Their Thallium(III) and Gold(III) Precursors

**DOI:** 10.3390/molecules28010447

**Published:** 2023-01-03

**Authors:** Laura Coconubo-Guio, José M. López-de-Luzuriaga, Sonia Moreno, M. Elena Olmos

**Affiliations:** Departamento de Química, Centro de Investigación en Síntesis Química (CISQ), Complejo Científico Tecnológico, Universidad de La Rioja, Madre de Dios 53, 26006 Logroño, Spain

**Keywords:** organometallic compounds, thallium(III), gold(III), perhaloaryl

## Abstract

In this paper, we describe a series of diphenylphosphane and diphenylphosphanide gold(III) and gold(III)/gold(I) complexes containing 3,5-C_6_Cl_2_F_3_ as aryl ligands at gold that have been synthesized due to the arylating and oxidant properties of the new polymeric thallium(III) complex [TlCl(3,5-C_6_Cl_2_F_3_)_2_]_n_ (**1**). Its reaction with [Au(3,5-C_6_Cl_2_F_3_)(tht)] (tht = tetrahydrothiophene) produces the gold(III) complex [Au(3,5-C_6_Cl_2_F_3_)_3_(tht)] (**2**), which allows the synthesis of the diphenylohosphane derivative [Au(3,5-C_6_Cl_2_F_3_)_3_(PPh_2_H)] (**3**). Its treatment with acetylacetonate gold(I) derivatives leads to two novel Au^III^/Au^I^ phosphanido-bridged complexes, [PPN][Au(3,5-C_6_Cl_2_F_3_)_3_(µ-PPh_2_)AuCl] (**4**) and [PPN][{(3,5-C_6_Cl_2_F_3_)_3_Au(µ-PPh_2_)}_2_Au] (**5**). All these complexes have been characterized, and the crystal structures of **1**, **2**, **4** and **5** have been established by single crystal X-ray diffraction methods, showing a novel polymeric arrangement in **1**.

## 1. Introduction

As is well known, the chemistry of gold(I) has been traditionally much more developed than that of gold(III), mainly due to the lower stability of this last oxidation number, at least with typical ligands in coordination chemistry, such as tertiary phosphanes, amines or halogens. Similarly, there is an even greater difference in the stability of the oxidation numbers +1 and +3 in the case of thallium, where the inert-pair effect, a concept that was first proposed by Sidgwick in 1927 [1], plays an important role. The number of monovalent thallium or gold derivatives is much greater than that of their trivalent congeners, and the stabilizing effect of perhalophenyl groups, more specifically, of C_6_F_5_, is evident in gold, where their use has facilitated the synthesis of trivalent gold compounds [2]. In the case of thallium, its chemistry has, in general, attracted less interest, and the advance of its organometallic chemistry is closely connected with the application of thallium intermediates in organic synthesis [3]. Furthermore, the coordination chemistry or organothallium complexes have been limited by their weak Lewis acid character (only about 70 structures of diorganothallium compounds had been reported prior to 1998) [4]. It is worth noting that, while a search in the CSD brings up 652 crystal structures of perhalophenylgold compounds (517 of them containing C_6_F_5_ as aryl group), only 9 structures of perhalophenythallium complexes (from which only one contains C_6_Cl_5_ instead of C_6_F_5_ [5]) are found. One of such compounds, specifically [Tl(C_6_F_5_)_2_Cl], was the key for the synthesis of the triarylgold(III) derivative [Au(C_6_F_5_)_3_(tht) [6], sometimes employed as starting material in gold(III) chemistry.

On the other hand, although tertiary phosphanes have been widely employed as ligands in coordination chemistry, secondary (and especially primary) phosphanes or their phosphanido derivatives have been used to a lesser extent, due to their much lower stability. In the particular case of gold, there is an infra-undergrowth in comparison with other transition metals such as palladium or platinum [7,8], in spite of the possible catalytic activity of di- and polynuclear phosphanido complexes [8,9,10,11]. In addition, PR_2_^−^ groups possess a strong nucleophilicity and a high bridging tendency, which could facilitate the presence of aurophilic interactions, frequently responsible for optical properties such as luminescence. As expected from the comments above regarding the relative stability of the oxidation numbers +1 and +3 in gold, only about the 10% of the gold phosphanido compounds reported contain gold(III) centers [12]. To the best of our knowledge, with the sole exceptions of [Au_2_Me_4_(µ-PPh_2_)_2_], reported by Puddephatt in 1976 [13], and [Au_2_{2,6-NC_5_H_3_(C_6_H_4_)_2_}_2_(µ-PPh_2_)]ClO_4_, published in 2008 [14], the remainder of gold(III) phosphanido gold(III) derivatives were published by our research group between 1998 and 2006 [15,16,17,18,19,20], all of them containing perfluorophenyl groups as stabilizing ligands at gold. Therefore, this area of the chemistry has not witnessed much progress over the last three lustra, with only about one paper per year on gold(I) phosphanido complexes, and the sole contribution on gold(III) derivatives by Li et al. [14].

With these precedents, we focused our interest in the synthesis of phosphanido gold(III) complexes, and with this aim, we thought of the possibility of employing the poorly represented 3,5-dichlorotrifluorophenyl group as a ligand bonded to gold, which could modify the electronic and steric effects in the new complexes and could confer them different stability and properties. However, surprisingly, the synthesis of gold(III) species with this perhalophenyl ligand seems to be not as obvious as expected, and when the same procedures as those employed to prepare *trans*-[Au(C_6_F_5_)_2 × 2_]^−^ [21], *cis*-[Au(C_6_F_5_)_2_X_2_]^−^ [21] and [Au(μ-Cl)(C_6_F_5_)_2_]_2_ [22] are followed, a quick rearrangement of aryl groups occurs, and a mixture of [Au(3,5-C_6_Cl_2_F_3_)_3_(solv)], [Au(3,5-C_6_Cl_2_F_3_)_2_Cl(solv)] and [Au(3,5-C_6_Cl_2_F_3_)Cl_2_(solv)] (solv = OEt_2_, OH_2_) is obtained [23].

For these reasons, we directed our efforts to the obtention of pure 3,5-dichlorotrifluorophenyl gold(III) derivatives containing diphenylphosphane or diphenylphosphanide as ligands. To achieve this goal, we first needed to prepare the appropriate starting materials: in an initial step, the thallium(III) derivative [Tl(3,5-C_6_Cl_2_F_3_)_2_Cl]_n_ (**1**) and, subsequently, the gold(III) complex [Au(3,5-C_6_Cl_2_F_3_)_3_(tht)] (**2**). Further treatment of **2** with PPh_2_H produced the gold(III) phosphane derivative [Au(C_6_Cl_2_F_3_)_3_(PPh_2_H)] (**3**), which was subsequently treated with acetylacetonate gold(I) compounds, following the “acac method” [24], to obtain mixed Au^III^/Au^I^ complexes containing bridging diphenylphosphanide ligands. The new compounds have been characterized by analytical and spectroscopic techniques, and the crystal structures of almost all of them have been determined by X-ray diffraction methods.

## 2. Results and Discussion

### 2.1. Synthesis and Characterization of the Thallium(III) Complex [Tl(3,5-C_6_Cl_2_F_3_)_2_Cl]_n_ (**1**)

#### 2.1.1. Synthesis and Spectroscopic Characterization of [Tl(3,5-C_6_Cl_2_F_3_)_2_Cl]_n_ (**1**)

As mentioned above, our interest was focused not only on the synthesis of new diphenylphosphane and phosphanido-bridged compounds, but also on the preparation of unknown starting materials for the subsequent synthesis of such gold(III) species. Considering the difficulties involved in the oxidation of the bis(aryl)aurate(I) anion [Au(3,5-C_6_Cl_2_F_3_)_2_]^−^ commented on above, which leads to mixtures of different aryl gold(III) complexes [23], we decided to employ the neutral gold(I) compound [Au(3,5-C_6_Cl_2_F_3_)(tht)] as starting material. Nevertheless, the first target was the synthesis of the appropriate oxidizing complex to treat [Au(3,5-C_6_Cl_2_F_3_)(tht)] with. Therefore, we first synthesized the novel aryl thallium(III) derivative [Tl(3,5-C_6_Cl_2_F_3_)_2_Cl]_n_ (**1**), which would supposedly act as oxidant and arylating agent if it behaved similarly to its pentafluorophenyl congener [6]. As expected, complex **1** was obtained as an air- and moisture-stable white solid by the transmetallation reaction that takes place between anhydrous thallium(III) chloride and two equivalents of a freshly prepared solution of 3,5-dichlorotrifluorophenyllithium in diethyl ether at −75 °C and under a nitrogen atmosphere (Figure 1). Its analytical and spectroscopic data are in accordance with the proposed stoichiometry and it behaves as a non-conducting species in acetone solutions.

The IR spectrum of **1** shows, among other features, absorption bands due to the presence of the aryl groups located at 1597, 1074 and 783 cm^−1^. Its ^19^F NMR spectrum in DMSO-*d_6_* shows two resonances, at −110.49 and −92.99 ppm, with a relative integration of 1:2, as expected for the inequivalent fluorine atoms in the *para* and *ortho* positions of the aryl rings, respectively. As a consequence of the coordination of the aryl ligands to thallium, each signal splits into doublets due to the spin–spin coupling between ^205^Tl and ^19^F, with Tl-F coupling constants of 759.9 (*^3^J_Tl-F__o_*) and 62.4 Hz (*^5^J_Tl-F__p_*), consistent with those described for other polyhalophenyl groups bonded to thallium [25]. Finally, the HRMS (+) of **1** shows a peak at *m/z* = 602.8369, corresponding to the cationic fragment [Tl(C_6_Cl_2_F_3_)_2_]^+^, with an experimental isotopic distribution in agreement with the calculated one.

#### 2.1.2. Crystal Structure of [Tl(3,5-C_6_Cl_2_F_3_)_2_Cl]_n_ (**1**)

It is worth mentioning that, although some complexes of the type [Tl(aryl)_2_Cl] (aryl = C_6_Cl_5_ [26], C_6_F_5_, 2,3,5,6-C_6_F_4_H, 2,3,4,6-C_6_F_4_H, 2,4,6-C_6_F_3_H_2_, 4-C_6_FH_4_, 3-CF_3_C_6_H_4_, [27] or mesityl [28]) had already been reported years ago, none of them had been studied by X-ray diffraction studies, and a dimeric structure with bridging chlorine atoms was supposed for them. We have determined the crystal structure of **1** by X-ray diffraction methods from single crystals obtained by slow diffusion of diethyl ether into a concentrated dichloromethane solution of the complex. Compound **1** crystallizes in the *P*2_1_/c space group of the monoclinic system, with one [Tl(3,5-C_6_Cl_2_F_3_)_2_Cl] fragment in the asymmetric unit with Tl–C bond lengths of 2.135(5) and 2.141(5) Å and a Tl–Cl distance of 2.6893(11) Å. The marked tendency of thallium to display high coordination numbers [4] together with the ability of chlorine to act as bridging ligand, which in this case connects three metal centers, result in the formation of an infinite polymeric zig-zag band of [Tl(3,5-C_6_Cl_2_F_3_)_2_Cl] units (parallel to the crystallographic *b* axis) with Tl–Cl bonds of similar length, (2.6893(11) and 2.7397(13) Å), which are tied together by somewhat longer Tl–Cl bonds (2.9856(13) Å), as shown in Figure 1. This structure somehow resembles the “dimeric” structure (really a polymer) described for [Tl(2,3,5,6-C_6_F_4_H)_2_Br] [29], although in our case no “dimers” are present.

The Tl–Cl distances within the zig-zag band [2.6893(11) and 2.7397(13) Å] are intermediate between those found for terminal Tl–Cl bonds in TlCl_3_·3C_5_H_5_N (2.520(6) and 2.498(4) Å) [30], [Tl(FcN)_2_Cl] (2.581(3) Å) [31], [Tl(C_6_F_5_)_2_Cl{Au(C_6_F_5_)_3_(OdppmO)}_2_] (2.469(7) Å) [32] or [Tl(C_6_F_5_)_2_(OdppmO)_2_][Tl(C_6_F_5_)_2_Cl_2_] (2.497(27) and 2.567(2) Å) [33], and those described for bridging Tl–Cl bond lengths in [TlMe_2_Cl]_n_ (3.029 Å) [34,35], [Tl(*p*-C_6_F_4_H)_2_Cl(OPPh_3_)]_2_ (2.936(3) Å) [36], {[Tl(8-Me_2_NC_10_H_6_)_2_]Cl}_n_ (2.929(2) and 2.933(2) Å) [37] or [TlPh_2_Cl(py)_2_]_n_ (2.907(4), 2.9285(5) and 2.951(4) Å) [38], being closer to the Tl–Cl distance found in {1,2-(Ph_2_TlN_4_C)_2_C_6_H_4_·Ph_2_TlCl·2MeOH·2H_2_O}_n_ (2.777(1) Å) [39], where the chlorine atom acts as a linear and symmetric bridge between two thallium centers, or to the shorter of the Tl–Cl lengths in [Tl(Me_3_SiCH_2_)_2_Cl]_2_ (2.76(1) and 2.99(1) Å) [40], which shows asymmetric Tl–Cl bond lengths. In fact, the latter is similar to the third Tl–Cl distance in the unidimensional polymer in **1** (2.9856(13) Å), which also compares well with most of the bridging Tl–Cl distances commented on above [34,35,36,37,38] (see Table 1).

An additional weaker Tl···Cl contact of 3.4716(11) Å that involves a chlorine atom of a perhaloaryl group appears between neighboring chains. It is worth noting that this weakly coordinated Cl is not a chloride, but part of a perhaloaryl group, which could, therefore, be seen as a sort of bridging ligand. This Tl···Cl contact, together with a couple of Cl···F interactions, [d(Cl-F) = 3.206(3) and 2.975(4) Å], give rise to an extended layer normal to the crystallographic *a* axis (see Appendix A). Finally, these layers are connected through pairs of Cl···F contacts of 3.193(4) Å, resulting in a 3D network (Appendix A).

Upon examination, the thallium environment, as can be seen in Figure 2, could be described as a very distorted octahedral one, where thallium is fairly [5+1] coordinated, with the four chlorine atoms lying in the same plane as the metal center and the carbon atoms of the aryl groups “perpendicular” to this plane (C1-Tl-C7 152.42(18)°). This strong deviation in the C-Tl-C angle, which bends toward the side of the weakest donor (the aryl-bound Cl ligand), is likely to be caused by the formation of the above-mentioned Tl···Cl_aryl_ and Cl···F interactions between aryl groups of neighboring chains (Appendix A).

### 2.2. Synthesis and Characterization of Gold(III) Complexes

#### 2.2.1. Synthesis and Spectroscopic Characterization of Gold(III) Complexes

The next step was the synthesis of the appropriate gold(III) precursor for the further preparation of phosphane and phosphanide complexes. As we expected, the thallium(IIII) derivative [Tl(3,5-C_6_Cl_2_F_3_)_2_Cl]_n_ (**1**) is capable of transferring its two aryl groups to gold(I) when treated with equivalent amounts of [Au(3,5-C_6_Cl_2_F_3_)(tht)] in toluene and under reflux. This oxidative addition reaction, which takes place with the simultaneous precipitation of thallium(I) chloride, leads to the synthesis of [Au(3,5-C_6_Cl_2_F_3_)_3_(tht)] (**2**), which is obtained as a white stable solid (Figure 2) whose analytical and spectroscopic data agree with the proposed stoichiometry. It is nonconducting in acetone, soluble in most of the common organic solvents and non-soluble in hexane.

The labile tetrahydrothiophene ligand present in **2** can be easily replaced by diphenylphosphane. Thus, reaction of **2** with PPh_2_H (1:1) in dichloromethane and under a nitrogen atmosphere leads to [Au(3,5-C_6_Cl_2_F_3_)_3_(PPh_2_H)] (**3**), one of the very few examples of gold complexes containing a primary or a secondary phosphane as ligand, obtained as an oily and sticky compound. All attempts to isolate it as a pure solid were unsuccessful; however, its spectroscopic data agree with the proposed stoichiometry for this compound and it was employed in situ, without isolating it, for subsequent reactions.

The IR spectrum of complex **2** shows, among other features, very strong absorptions arising from the C_6_Cl_2_F_3_^−^ groups at 1600 (vs), 1062 (vs) and 781 (vs) cm^−1^, as well as a couple of weaker bands at 1310 and 1250 cm^−1^, which correspond to the sulfur donor ligand.

The ^19^F NMR spectrum of **2** displays two groups of resonances at −94.94 and −95.41 ppm (due to 2 and 4 F*_o_* atoms, respectively), and at −112.51 and −113.57 ppm (due to 2 and 1 F*_p_* atoms, respectively), as expected for a complex with 2 types of inequivalent 3,5-dichlorotrifluorophenyl rings bonded to gold(III). The ^19^F NMR spectrum of compound **3** is very similar to that of **2**, showing only small shifts in the location of the signals [−94.71 (4 F*o*), −95.08 (2 F*o*), −114.07 (2 F*p*) and −114.55 ppm (1 F*p*)]. In addition, the signals of the F*_o_* atoms appear broadened in both cases because of their coupling with other fluorine atoms of the aryl ligands as well as with the phosphorus of PPh_2_H in **3**.

Regarding their ^1^H NMR spectra, that of complex **2** shows only 2 multiplets centered at 3.14 and 2.05 ppm, with relative intensities of 1:1, corresponding to the methylenic protons of the tetrahydrothiophene ligand. In the case of **3**, its ^1^H NMR spectrum, in addition to the signals attributed to the aromatic protons between 7.8 and 7.4 ppm, displays a resonance centered at 6.18 ppm, corresponding to the highly unshielded hydrogen atom directly bonded to phosphorus, which, therefore, appears as a doublet with a large P-H coupling constant [*J*_P-H_ = 414.1 Hz]. The presence of the secondary phosphane is also revealed by its ^31^P{^1^H} NMR spectrum, which displays a broad signal (because of the coupling of the phosphorus with the *ortho* fluorine atoms of the aryl group located in *trans* to PPh_2_H) at −8.38 ppm that is shifted to low field relative to the resonance of the free ligand, which appears at −40.3 ppm, as a consequence of its coordination with gold.

Finally, the HRMS (−) spectrum of **3** shows that the peak corresponding to the anion [Au(C_6_Cl_2_F_3_)_3_(PPh_2_)]^−^ was found at 978.8187 Da, with an experimental isotopic distribution in agreement with the calculated one.

#### 2.2.2. Crystal Structure of [Au(3,5-C_6_F_5_)_3_(tht)] (**2**)

The molecular structure of **2** was determined by X-ray diffraction methods from single crystals grown by slow evaporation of a hexane solution of the complex, which crystallizes in the monoclinic *P*2_1_/n space group. Its molecular structure consists of [Au(3,5-C_6_F_5_)_3_(tht)] neutral molecules in which the square-planar gold(III) center is surrounded by three C atoms of the dichlorotrifluorophenyl groups and the sulfur donor atom of a tetrahydrothiophene molecule (Figure 3).

The mutually *trans* Au–C bond lengths, of 2.073(3) and 2.068(3) Å, are longer than the Au–C *trans* to sulfur, of 2.035(3) Å, according to the higher *trans* influence of the aryl groups than S-donor ligands, and compare well with the Au–C bond lengths described for the mutually *trans* aryl groups in [Au(3,5-C_6_Cl_2_F_3_)_3_(OH_2_)] (2.056(7) Å) or in the anion [Au(3,5-C_6_Cl_2_F_3_)_4_]^−^ (2.059(5) Å) [23] (Table 2). In addition, all the Au–C bond lengths in [Au(3,5-C_6_Cl_2_F_3_)_3_(tht)] (**2**) are almost equal to those found for its pentafluorophenyl analogue (Au–C *trans* to C_6_F_5_: 2.056(5)-2.068(2) Å and Au–C *trans* to S: 2.030(5) and 2.032(3) Å) [41]. In contrast, the Au–S bond length of 2.3897(8) Å is slightly longer than those found in [Au(C_6_F_5_)_3_(tht)] (2.3623(13) and 2.37792(9) Å) [41], and surprisingly longer than the Au^I^-S bond lengths found in [Au(C_6_F_5_)(tht)]_n_ (2.317(3) and 2.320(3) Å) [41] and [Ag_2_Au(3,5-C_6_Cl_2_F_3_)(CF_3_CO_2_)_2_(tht)]*_n_* (2.3158(16) Å) [42], which contain gold(I) instead of gold(III) centers in the linear [Au(C_6_X_5_)(tht)] units (see Table 2).

No apparent reason for this fact is found in the supramolecular structure of **2**, where neither the sulfur atom nor the gold one participates in weak interactions. C–H···F hydrogen bonds and weak Cl···Cl contacts between adjacent molecules result in a 3D array instead (see Appendix A).

### 2.3. Synthesis and Characterization of Gold(III)/Gold(I) Phosphanido Complexes

#### 2.3.1. Synthesis and Spectroscopic Characterization of Gold(III)/Gold(I) Phosphanido Complexes

As commented on in the introduction, our goal was to obtain mixed Au^III^/Au^I^ complexes containing bridging diphenylphosphanide ligands and, with this aim, we explored the use of [Au(3,5-C_6_Cl_2_F_3_)_3_(PPh_2_H)] (**3**) as starting material. Considering the deprotonating ability of acetylacetonate [24], which is able to deprotonate primary or secondary phosphane ligands [12], we treated a freshly prepared solution of **3** in dichloromethane with the acetylacetonate gold(I) derivates [PPN][Au(acac)Cl] (PPN = [N(PPh_3_)_2_]^+^) or [PPN][Au(acac)_2_] in 1:1 or 2:1 molar ratios, respectively. As expected, these reactions take place with the generation of diphenylphosphanide gold(III) units that rapidly incorporate the gold(I) fragments to the phosphorus atoms, thus producing the di- and trinuclear Au^III^/Au^I^ phosphanido-bridged gold species [PPN][Au(3,5-C_6_Cl_2_F_3_)_3_(µ-PPh_2_)AuCl] (**4**) and [PPN][{(3,5-C_6_Cl_2_F_3_)_3_Au(µ-PPh_2_)}_2_Au] (**5**), respectively, with the concomitant formation of acetylacetone (see Figure 3). It is worth noting that, as far as we are aware, these are the first dichlorotrifluorophenyl phosphanido complexes ever reported.

Both compounds **4** and **5** are obtained as white solids that behave as 1:1 electrolyte in acetone solutions, and their analytical and spectroscopic data agree with the proposed formulation for these complexes. Their IR spectra display, among other features, absorptions assigned to the PPN^+^ cation (at 534 cm^−1^) as well as to the aryl groups bonded to gold(III) at about 1590, 1055 and 781 cm^−1^. In addition, a weak band at 322 cm^−1^ appears in the IR spectrum of **4**, which corresponds to the Au^I^֪–Cl bond.

The ^19^F NMR spectra of **4** and **5** in CDCl_3_ are nearly identical and display the same pattern as that observed in the spectra of complexes **2** and **3**, although with the resonances slightly displaced, confirming the presence of the tris(aryl)gold(III) fragments. On the other hand, their ^31^P{^1^H} NMR spectra show a sharp singlet at 21.1 ppm, confirming the presence of the cation PPN^+^, and a broader resonance due to the phosphorus of the PPh_2_^−^ ligands, which appear at 19.48 (**4**) or 30.9 ppm (**5**), that is, low-field shifted when compared with that of **3** (−8.38 ppm).

Regarding their ^1^H NMR spectra, the disappearance of the doublet assigned to the PPh_2_*H* hydrogen in the ^1^H NMR spectra of **3** confirms the absence of H–P bonds, displaying only signals assigned to aromatic hydrogen atoms.

Finally, in their HRMS (−), the peaks corresponding to the molecular anion [Au(C_6_Cl_2_F_3_)_3_(µ-PPh_2_)AuCl]^−^ (**4**) or [{(C_6_Cl_2_F_3_)_3_Au(µ-PPh_2_)}_2_Au]^−^ (**5**) appear at *m/z* = 1210.7519 or 2154.6073, respectively, with experimental isotopic distributions in agreement with the calculated ones.

#### 2.3.2. Crystal Structures of PPN[Au(3,5-C_6_Cl_2_F_3_)_3_(µ-PPh_2_)AuCl] (**4**) and PPN[{(3,5-C_6_Cl_2_F_3_)_3_Au(µ-PPh_2_)}_2_Au] (**5**)

Crystals of **4** and **5** suitable for X-ray diffraction studies were obtained from solutions of the corresponding complex in tetrahydrofuran layered with hexane. The molecular structure of complex **4** displays PPN^+^ cations and [Au(3,5-C_6_Cl_2_F_3_)_3_(µ-PPh_2_)AuCl]^−^ anions, the latter containing a diphenylphosphanide fragment that connects a tris(aryl)gold(III) unit and a chloroaurate(I) anion (Figure 4), while in the case of complex **5**, the molecular anion is formed by two tris(aryl)diphenylphosphanideaurate(III) units linked by a linear gold(I) atom, which lies in the inversion center (Figure 5). The gold(I) atom in **4** is also linearly coordinated, with a deviation of the ideal angle of only 1.54°.

The [Au(3,5-C_6_Cl_2_F_3_)_3_(µ-PPh_2_)]^−^ fragments are essentially equal in both structures showing essentially square-planar geometries for the gold(III) atoms, with the Au^III^ center (Au1) lying on the plane defined by the *ipso* carbon atoms of the three aryl groups (C1, C7 and C13) and the phosphorus atom (P1) at 0.617 (**4**) or at 0.256 Å (**5**) of the plane.

The Au^III^-Au^I^ distance, of 3.9483(5) Å in **4** and 3.9391(3) Å in **5**, is too large to consider any kind of gold···gold interaction. The Au–C bond lengths (Table 3), in the range 2.058(5)–2.076(5) Å in **4** and 2.044(5)–2.066(5)Å in **5**, compare well with those observed for the mutually *trans* carbons in **2** and also with those reported for related tris(pentafluorophenyl)gold(III) phosphanide compounds [15,16,17,19,20]. The similarity of the three Au–C bond lengths in each complex evidences a similar *trans* influence for dichlorotrifluorophenyl and diphenylphosphanide groups, both of them higher than the *trans* influence of the S-donor ligand. Taking this *trans* influence order into account, one would expect to find similar Au–P distances to those located in *trans* to an aryl group or to phosphorus, or even shorter for those involving gold(III) centers, according to the smaller radius expected for Au^III^ than for Au^I^. In contrast, the Au^III^–P (*trans* to 3,5-C_6_Cl_2_F_3_) bond lengths in **4** (2.3707(13) Å) and in **5** (2.3645(11) Å) are longer than the Au^I^–P (*trans* to PPh_2_) in **5** (2.3079(13) Å), and all of them longer than the Au^I^–P length in **4** (2.2436(13) Å), where the phosphorus is located in *trans* to a chlorine atom, with lower *trans* influence. Similar Au^III^–P and Au^I^–P bond lengths have been reported for other pentafluorophenyl gold(III) phosphanido complexes [15,16,17,18,19,20]. Regarding the Au^I^-Cl bond length in **4**, of 2.2830(16) Å, it is shorter than those found in the dinuclear gold(I) phosphanides PPN[(AuCl)_2_(μ-PR_2_)] (R = t-Bu, cyclo-C_6_H_11_), which are 2.3027(14) and 2.3083(13) Å (R = cyclo-C_6_H_11_) and 2.3141(10) and 2.3140(11) Å (R = t-Bu) [43].

Finally, intermolecular C–H···F and C–H···Cl hydrogen bonds between the counterions give rise to an extended supramolecular one-dimensional array in **4** (Appendix A), while in the case of complex **5**, only intramolecular C–H···F interactions are observed within the molecular anion (Appendix A).

## 3. Experimental Section

### 3.1. General

All reactions were performed in oven-dried glassware with magnetic stirring under a deoxygenated nitrogen atmosphere. The starting complexes [Au(3,5-C_6_Cl_2_F_3_)(tht)] (tht = tetrahydrothiophene) [44], [PPN][Au(acac)Cl] [45] and [PPN][Au(acac)_2_] [45,46] were prepared as described in the literature. The starting materials were obtained from commercial sources; these included 1,3,5-tricloro-2,4,6-trifluorobenzene (Fluorochem, Old Glossop, UK and n-Butyllithium solution 1.6 M in hexane (Sigma-Aldrich, Madrid, Spain), and were used as received. PPh_2_H was purchased from Sigma-Aldrich and used as a 0.96 M tetrahydrothiophene solution. Thallium(III) chloride was purchased from Sigma-Aldrich and anhydrous thallium(III) chloride was prepared according to the known procedure as described in the literature [47]. Because of the toxic nature of many thallium compounds, rubber gloves were worn throughout this portion of the experimental study. All solvents used for the synthesis of the new compounds were obtained from commercial sources and were used without further purification.

### 3.2. Instrumentation

Infrared spectra were recorded in the 4000–225 cm^−1^ range on a Nicolet Nexos FT-IR Spectrum (Thermo Nicolet Corporation, Madison, WI, USA) using Nujol mulls between polyethylene sheets. Conductivities were measured in ca. 5 × 10^−4^ M acetone solutions with a Jenway 4010 conductimeter (Jenway, Felsted, UK). High resolution mass spectrometry (HRMS) was performed using a time-of-flight mass spectrometer equipped with an ESI ionization source (Bruker MicroTOF-Q spectrometer, Bruker Corporation, Bremen, Germany). The analyses were carried out in negative and positive mode. ^31^P{^1^H}, ^19^F and ^1^H NMR experiments were recorded on a Bruker AVANCE 400 (Bruker Corporation, Fällanden, Switzerland) in CDCl_3_ solutions or on a Bruker ARX300 in CDCl_3_ or DMSO-*d_6_* solutions. Chemical shifts are quoted relative to H_3_PO_4_ (^31^P, external), CFCl_3_(^19^F, external) and SiMe_4_ (^1^H, external).

### 3.3. Synthesis

#### 3.3.1. Synthesis of [Tl(3,5-C_6_Cl_2_F_3_)_2_Cl]_n_ (**1**)

A round-bottom 2-neck flask was cooled to −75 °C and charged with 1,3,5-trichloro-2,4,6-trifluorobenzene (3.5004 g, 14.869 mmol) in 70 mL diethyl ether, whereupon a solution of butyllithium in diethyl ether (9.2931 mL, 14.869 mmol) was added dropwise. As soon as the addition was completed, the dropping funnel was replaced by a mineral oil bubbler and the mixture was stirred under a nitrogen atmosphere for 4 h and 30 min while the temperature was maintained at −75 °C.

Anhydrous thallium trichloride (2.3000 g, 7.4017 mmol) was added to the above solution of (3,5-dichlorotrifluorophenyl)lithium and the mixture was stirred for 1 h and 30 min at −75 °C, after which it was allowed to reach room temperature (slowly ~1 h). The nitrogen stream was stopped, and the precipitated lithium chloride was removed by filtration over celite. The yellow-orange filtrate was treated with 80 mL water previously acidified with 4 drops of concentrated HCl (addition of water followed by acidification leads to OH^−^ containing end products). Then, the organic layer—which, during this operation, has lost color—was removed by using a separating funnel, and the aqueous layer—which was extracted with diethyl ether (3 × 10 mL)—were then added to the previously separated diethyl ether solution and dried over 1–2 g anhydrous magnesium sulfate.

After filtration, the diethyl ether was removed at reduced pressure, and cold hexane (20 mL) was added to produce a white precipitate. The mixture was stirred for 5 min, and the product was isolated by filtration, washed with cold hexane (3 × 1 mL) and dried under vacuum to produce the new complex [Tl(3,5-C_6_Cl_2_F_3_)_2_Cl]_n_ (**1**) as a white solid (2.2896 g, 3.5788 mmol, 53%).

*Λ_M_* (acetone): 5.0 Ω^−1^ cm^2^ mol^−1^. HRMS (+): *m/z* = 602.8369 Da, [Tl(C_6_Cl_2_F_3_)_2_]^+^ (calculated = 602.839709 Da). IR: ν(C_6_Cl_2_F_3_) = 1597 (vs), 1074 (vs) and 783 (vs) cm^−1^; ^19^F NMR (DMSO-d_6_, 298K), δ: −92.99 (d, 4F, F*_o_*, *^3^J_Tl-F_* = 759.9 Hz), −110.49 ppm (d, 2F, F*_p_*, *^5^J_Tl-F_* = 62.4 Hz).

#### 3.3.2. Synthesis of [Au(3,5-C_6_Cl_2_F_3_)_3_(tht)] (**2**)

[Au(3,5-C_6_Cl_2_F_3_)(tht)] (0.4851 g, 1.000 mmol) was added to a solution of [Tl(3,5-C_6_Cl_2_F_3_)_2_Cl]_n_ (**1**) (0.6400 g, 1.000 mmol) in 30 mL of toluene, and after a few minutes, the solution turned yellowish as solid thallium chloride started to appear. After 7 h stirring at room temperature followed by 14 h refluxing, and after allowing the solution to reach room temperature, the TlCl that had completely precipitated was filtered off and the filtrate evaporated to dryness in vacuo. Hexane (20 mL) was then added, and a white precipitate started to arise. The mixture was stirred for 5 min; the solid was separated by filtration, washed with hexane (3 × 1 mL) and dried under vacuum to produce [Au(3,5-C_6_Cl_2_F_3_)(tht)] (**2**) as a white solid (0.4956 g, 0.6303 mmol, 56%).

*Λ_M_* (acetone): 1.0 Ω^−1^ cm^2^ mol^−1^. IR: ν(C_6_Cl_2_F_3_) = 1600 (vs), 1062 (vs) and 781 (vs) cm^−1^; ν(SC_4_H_8_) = 1310 (w) and 1250 (w) cm^−1^; ^1^H NMR (CDCl_3_, 298K), δ: 3.14 (m, 4H, S-CH_2_-CH_2_), 2.05 (m, 4H, S-CH_2_-CH_2_). ^19^F NMR (CDCl_3_, 298K), δ: -94.94 (m, 2F, F*_o_*), −95.41 (m, 4F, F*_o_*), −112.51 (s, 2F, F*_p_*), −113.57 (s, 1F, F*_p_*).

#### 3.3.3. Synthesis of [Au(3,5-C_6_Cl_2_F_3_)_3_(PPh_2_H)] (**3**)

PPh_2_H (0.20 mL, 0.20 mmol) was added to a solution of [Au(3,5-C_6_Cl_2_F_3_)_3_(tht)] (**2**) (0.1770 g, 0.2000 mmol) in 20 mL of diethyl ether under a nitrogen atmosphere. After 30 min of stirring, the solution was concentrated under vacuum to ca. 5 mL and, by addition of cold hexane (20 mL) while stirring the mixture, complex [Au(3,5-C_6_Cl_2_F_3_)_3_(PPh_2_H)] (**3**) was precipitated as a white oily and sticky material (0.1140 g, 0,1160 mmol, 58%).

HRMS (−): *m/z* = 978.8187 Da, [Au(C_6_Cl_2_F_3_)_3_(PPh_2_)]^−^ (calculated = 978.817874 Da); ^1^H NMR (CDCl_3_, 298K), δ: 7.8–7.4 (m, 10H, Ph), 6.18 (d, 1H, PPh_2_H, *J_P-H_* = 414.1 Hz); ^19^F NMR (CDCl_3_, 298K), δ: −94.71 (m, 4F, F*_o_*), −95.08 (m, 2F, F*_o_*), −114.07 (s, 2F, F*_p_*), −114.55 (s, 1F, F*_p_*); ^31^P{^1^H} NMR (CDCl_3_, 298K), δ: −8.38 (br s).

#### 3.3.4. Synthesis of [PPN][Au(3,5-C_6_Cl_2_F_3_)_3_(µ-PPh_2_)AuCl] (**4**)

To a freshly prepared solution of [Au(3,5-C_6_Cl_2_F_3_)_3_(PPh_2_H)] (**3**) (0.1966 g, 0.2000 mmol) in 20 mL of dichloromethane, [PPN][Au(acac)Cl] (0.1740 g, 0.2000 mmol) was added. After the reaction mixture was stirred for 2 h at room temperature, total evaporation of the solvent under vacuum and addition of cold hexane (20 mL) gave rise to the precipitation of a white solid. It was filtered, washed with cold hexane (3 × 1 mL) and dried under reduced pressure to produce [PPN][Au(3,5-C_6_Cl_2_F_3_)_3_(µ-PPh_2_)AuCl] (**4**) as a pure white solid in an almost quantitative yield (0.3435 g, 0.2213 mmol, 98%).

*Λ_M_* (acetone): 120 Ω^−1^ cm^2^ mol^−1^. HRMS (−): *m/z* = 1210.7519 Da, [Au(C_6_Cl_2_F_3_)_3_(µ-PPh_2_)AuCl]^−^ (calculated = 1210.753296 Da). HRMS (+): *m/z* = 538.1853 Da, [PPN]^+^ (calculated = 538.184800 Da). IR: ν(C_6_Cl_2_F_3_) = 1589 (vs), 1053 (vs) and 781 (vs) cm^−1^; ν(Au^I^-Cl) = 322 (w) cm^−1^; ν(PPN^+^) = 534 (s) cm^−1^. ^1^H NMR (CDCl_3_, 298K), δ: 7.66–7.45 (m, 10H, Ph); ^19^F NMR (CDCl_3_, 298K), δ: −92.11 (m, 4F, F*_o_*), −93.30 (m, 2F, F*_o_*), −117.82 (s, 2F, F*_p_*), −118.45 (s, 1F, F*_p_*). ^31^P{^1^H} NMR (CDCl_3_, 298K), δ: 19.48 (s, PPh_2_^−^), 21.14 (s, PPN^+^).

#### 3.3.5. Synthesis of [PPN][{(3,5-C_6_Cl_2_F_3_)_3_Au(µ-PPh_2_)}_2_Au] (**5**)

To a freshly prepared solution of [Au(3,5-C_6_Cl_2_F_3_)_3_(PPh_2_H)] (**3**) (0.1966 g, 0.200 mmol) in 20 mL of dichloromethane, [PPN][Au(acac)_2_] (0.0938 g, 0.1000 mmol) was added. After the reaction mixture was stirred for 2 h at room temperature, the solvent was concentrated to ca. 5 mL in vacuo. Cold hexane (20 mL) was added and the mixture was stirred for 5 min; the precipitate thus formed was separated by filtration, washed with hexane (3 × 1 mL) and dried under vacuum to produce [PPN][{(3,5-C_6_Cl_2_F_3_)_3_Au(µ-PPh_2_)}Au] (**5**) as a white solid (0.2698 g, 0.0999 mmol, 50%).

*Λ_M_* (acetone): 106 Ω^−1^ cm^2^ mol^−1^. HRMS (−): *m/z* = 2154.6073 Da, [{(3,5-C_6_Cl_2_F_3_)_3_Au(µ-PPh_2_)}Au]^−^ (calculated = 2154.601769 Da). HRMS (+): *m/z* = 538.1850 Da, [PPN]^+^ (calculated = 538.184800 Da). IR: ν(C_6_Cl_2_F_3_) = 1591 (vs), 1055 (vs) and 781 (vs) cm^−1^; ν(PPN^+^) = 534 (s) cm^−1^. ^1^H NMR (CDCl_3_, 298K), δ: 7.67–7.44 (m, 10H, Ph); ^19^F NMR (CDCl_3_, 298K), δ: −92.14 (m, 8F, F*_o_*), −93.33 (m, 4F, F*_o_*), −117.84 (s, 4F, F*_p_*), −118.48 (s, 2F, F*_p_*); ^31^P{^1^H} NMR (CDCl_3_, 298K), δ: 30.9 (s, PPh_2_^−^), 21.10 (s, PPN^+^).

### 3.4. Crystallography

Crystals were mounted in inert oil on glass fibers and transferred to the cold gas stream of a Bruker APEX-II CCD diffractometer equipped with an Oxford Instruments low-temperature attachment controller system (Mo Kα = 0.71073 Å, graphite monochromator). Data were collected in ω- and φ-scan modes. Absorption effects were treated by numerical correction. The structures were solved with the SHELXT 2014/5 program [48] and refined by a full-matrix least-squares procedure based on *F*2 with SHELXL 2018/3 [49] using Olex2 program suite [50]. All non-hydrogen atoms, except those of the PPN^+^ cation in **5**, which is highly disordered, were refined anisotropically. A complex model has been defined to solve this disorder, considering the phosphorus atom disordered over three positions with ratio 0.6:0.2:0.2, and the phenyl rings disordered over two (C31→C36 and C43→C48) or three positions (C37→C42), with ratios 0.7:0.3 or 0.7:0.15:0.15, respectively. Furthermore, an additional high residual electron density was found between ions in **5**, presumably due to heavily disordered solvent molecules, so the structure was further analyzed using the BYPASS subroutine implemented in Olex2. This analysis indicates the presence of 182 electrons per unit cell, which is consistent with the presence of 4 molecules of disordered THF (OC4H8) per asymmetric unit. Hydrogen atoms were placed in calculated positions refined using a riding model. Geometrical calculations were performed using PLATON [51] and the structure drawings were executed with MERCURY [52] programs. Table 4 contains crystallographic data and structure refinement details for compounds 1, 2, 4 and 5 and CCDC 2221966–2221969 contain the supplementary crystallographic data for this paper. These data can be obtained free of charge via www.ccdc.cam.ac.uk/conts/retrieving.html (accessed on 23 November 2022), or by contacting The Cambridge Crystallographic Data Centre, 12 Union Road, Cambridge CB2 1EZ, UK; fax: +44 1223 336033.

## 4. Conclusions

The novel diorganothallium(III) chloride [Tl(3,5-C_6_Cl_2_F_3_)_2_Cl]_n_ (**1**), which shows a three-dimensional structure formed due to Tl···Cl and Cl···F interactions of different intensities, has been demonstrated to be an effective oxidizing and arylating reagent against the gold(I) precursor [Au(3,5-C_6_Cl_2_F_3_)(tht)]. This synthetic route has allowed us to prepare one of the first pure dichlorotrilfluorophenylgold(III) complexes, [Au(3,5-C_6_Cl_2_F_3_)_3_(tht)] (**2**), since this type of poorly represented compound tends to produce mixtures of different species in equilibrium.

The aforementioned complex is the key to the synthesis of a series of gold(III) diphenylphosphane and -phosphanide derivatives aided by the stabilizing effect of perhalophenyl ligands. Thus, we have prepared the Au^III^ neutral complex [Au(3,5-C_6_Cl_2_F_3_)_3_(PPh_2_H)] (**3**), which is employed as starting material for the further synthesis of the mixed Au^III^/Au^I^ anionic phosphanide compounds [PPN][Au(3,5-C_6_Cl_2_F_3_)_3_(µ-PPh_2_)AuCl] (**4**) and [PPN][{(3,5-C_6_Cl_2_F_3_)_3_Au(µ-PPh_2_)}_2_Au] (**5**) by a simple reaction pathway that makes use of the deprotonating and coordinating ability of acetylacetonate gold complexes. These are among the scarce gold phosphanide compounds and represent the first examples of phosphane or phosphanide gold complexes with 3,5-dichlorotrifluorophenyl groups as ligands.

## Data Availability

Not applicable.

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
