# Peer review of "Synthesis and Structural Characterization of Phosphanide Gold(III)/Gold(I) Complexes and Their Thallium(III) and Gold(III) Precursors"

_molecules, 2023, doi:10.3390/molecules28010447_

Round 1

Reviewer 1 Report

The authors have prepared a new Tl(III)-compound TlCl(C6Cl2F3)2, which is capable of transferring the haloaryl groups onto Au(C6Cl2F3)(tht) with formation of TlCl and Au(C6Cl2F3)3(tht). They have shown that the (tht) ligand can be replaced by Ph2PH with formation of Au(C6Cl2F3)3(Ph2PH). The latter was used as a starting material in the syntheses of Ph2P-bridged Au(III)Au(I) complexes. Four out of the five new compounds were characterized crystallographically, and characterization of all of them includes IR, multi-nuclear NMR (1H, 19F, 31P) and MS as well as conductivity. The subject matter is suitable for this journal, and the experiments have certainly been carried out in a fine manner. In my opinion, characterization is fine. The presentation of the results, however, deserves some further effort. Therefore, I recommend publication upon major revision (which should take the following points into account, that´s a long list already, but I simply stopped at this point, the number of points listed should be sufficient to show that the presentation of the nice results requires a lot of polishing):

1)      In the abstract the authors write “fully characterized”. To be honest, I hate this expression. There is no such thing as full characterization (unless all, really all, methods available have been used), an expression like “comprehensive characterization” or “characterized with various methods” is what the authors mean to say and what is true. (For example, even in case of the methods applied the authors have not taken all options into consideration, 13C NMR instantly comes to my mind.)

2)      Fluoroaryl and perhaloaryl should be added to the key words (they are important parts of this paper and do not appear in the title).

3)      The writing style needs to be improved in some way. For example, in lines 27,28 the authors write “If one takes a look into the organometallic chemistry of these two metals…”, but this piece can/should be omitted, because the next words (“…the number of monovalent thallium or gold derivatives is enormously higher…”) are also true even if we don´t take a look (into the relevant literature, not into the chemistry, by the way). Another example: In lines 31-36 it reads “In the case of thallium, its chemistry has in general attracted less interest, and the advance of its organometallic chemistry is closely connected with the application of thallium intermediates in organic synthesis [3], and the coordination chemistry or organothallium complexes has been limited by their weak Lewis acid character (only about 70 structures of diorganothallium compounds had been reported prior to 1998) [4].” This is a pretty long sentence, which could easily be modified toward shorter sentences which express the same facts (which are not necessarily connected to one another). It might better read “In the case of thallium, its chemistry has in general attracted less interest, and the advance of its organometallic chemistry is closely connected with the application of thallium intermediates in organic synthesis [3]. Furthermore, the coordination chemistry of organothallium complexes has been limited by the weak Lewis acid character of their Tl(III) center (only about 70 structures of diorganothallium compounds had been reported prior to 1998) [4].” Another point (which I usually criticize) is the use of the expression “bond distance”. It is either bond length (length of a particular bond) or interatomic distance (distance between two atoms), whereas “bond distance” would mean the distance between two bonds. “Bonding distance” might be an alternative, but it bears the risk of being mistaken for the distance (in general) at which an interatomic contact is referred to being an actual bond. Line 99 “a non-conducting electrolyte” sounds to me like an oxymoron.

4)      Furthermore, various typos (e.g., line 110 “polyhalopenhyl”) need to be corrected, and some general polishing of language is recommended.

5)      Discussion of the TlCl(C6Cl2F3)2 structure: To me the structure does not reflect any dimeric character, it represents an infinite polymeric zig-zag band of TlCl(C6Cl2F3)2 units with Tl-Cl bonds of similar length (2.69 and 2.74 A), which are tied together by somewhat longer Tl-Cl bonds (2.99 A). Also, the surprisingly long Tl-Cl separation of 3.47 A needs to be discussed in the context of a chloroaryl group acting as a ligand (this weakly coordinating Cl is not a chloride, it is part of an aryl group, and therefore a ligand worth being highlighted. Hence, Tl is sort of [5+1]-coordinated. This is (in my opinion) the cause of the C-Tl-C angle bending (toward the side of the weakest donor, i.e., toward the aryl-bound Cl ligand). There is some room because of the distance of the sixth ligand atom.

6)      In Figure 1, the bonds Tl-Cl1#1 (looks long) and Tl-Cl1#2 (looks short) are probably mismatched in the figure caption OR in the labelling scheme. The short bond in the Figure caption looks like the short bond in the picture.

7)      For the crystal structures, the way of selecting atom labels should be mentioned in the figure caption OR all non-H atoms of the asymmetric unit should be labelled.

8)      Lines 285,286 “a unique perfectly linear gold(I) atom”? It is the coordination sphere (not the atom), which is linear. Is it unique? No, it is a symmetry-imposed linearity. Even if the bond angle at Au were something like 179 or maybe even 178 deg., we would not be able to see it in this crystal structure in which the Au atom is forced to sit on a center of inversion.

9)      Line 314: radius (not radium)

10)   In the general experimental section, sources of some of the chemicals used are not mentioned (BuLi, TlCl3, C6Cl3F3).

11)   Descriptions of syntheses needs to be a bit more detailed. “…is treated with cold hexane (20 mL)…”, and then there is a yield. Treated?...Hexane added for precipitation? Or washed with hexane maybe? Hexane added, stirred, filtered, and the washed maybe? To dry or not to dry, that´s the (next) question.

12)   In order to make calculations of yield easier, for the starting materials used the amounts used (g and mol) should be reported with the same precision (same number of significant places), and for the yield, g, mol and % should be reported. 0.4851 g, 1 mmol…is it exactly 1.000 mmol? If yes, then write 0.4851 g, 1.000 mmol or better use 3 significant digits, 0.485 g, 1.00 mmol.

13)   Crystallography: “and refined and refined” sounds good and convincing, but writing “and refined” is sufficient. Also, instead of SHELXL-97 other versions had been used (according to the CIFs).

14)   Ref. 6: One of the initials of author Jones is missing.

Reviewer 2 Report

The paper describes the synthesis of thalium(III) organometallic compounds containing polyhalogenated aryl rings sigma bonded to the metal via a carbon atom. This compound was then used as a key agent that could oxidatively transmetalate its polyhaloaryl onto a gold(I) organometallic species leading to a Gold(III) molecule from which a phosphane adduct was obtained that further lead to a mixed Au(III)/Au(I) complexes containing bridging diphenylphosphanide ligands. The goal of the work was to find reliable ways to the last mentionned molecular unit, and this has indeed been achieved. The work was competently performed, the various obtained compounds having been identified by different spectroscopic tools including by X-ray diffraction studies on single crystals.

The paper deserves to be published provided minor changes have been brought about :

The paragraphs (lines 140 - 150 and lines 224 - 234)that describes the structures of  (1) and (2) are quite difficult to read as they both contain many bond lengths and angles; they could be improved if these data would be gathered in a table that allows an easier comparaison of the structures therein reported.

Besides this there are some typing errors that can be easily corrected, examples are as follows :

line 66 : 'ligands seems to be not so obvious es expected' should read : 'ligands seems to be not as obvious as expected'; line 90, change 'compund' by 'compound'; line 92, the end of the sentence is missing; line 110, change ''polyhalopenhyl' for 'polyhalophenyl'; lines 160-161 'Regarding the thallium environment, as can be seen in Figure 2, it could be described as a very distorted octahedral one' could be changed for 'Examining the thallium environment, as can be seen in Figure 2, it could be described
as a very distorted geometry' one'; etc...

Reviewer 3 Report

The manuscript reports the interesting synthesis of new gold(III) and gold(III)\gold(I) derivatives. The starting materials for obtaining these gold complexes is a new thallium compound  [Tl(3,5-C6Cl2F3)2Cl]n (1), the X-ray  structure of which is correctly described. Besides this initial Tl complex, 3 gold X-ray structures are described: [Au(3,5-C6Cl2F3)3(tht)] (2) , PPN[Au(3,5-C6Cl2F3)3(μ-PPh2)AuCl] (4) and PPN[{(3,5-C6Cl2F3)3Au(μ-PPh2)}2Au].

If the synthetic work is well described, some important informations are missing from the X-ray. Indeed the major interest of the paper is the X-ray structure of new gold compounds and then the reaser will expect at least a crystal data table and maybe some tables containing intersting geometry compared with related gold complexes. If the structures of complexes 1,2 and 4 are well solved without difficulties, there are serious problem in compound 5.

-first of all the initial date have been squeezed certainly beacuse of the occurrence of tht disordered. But there is no mention concerning the use of SQUEEZE in the experimental section.

- The disordered PPN seems for me irrealistic and not chemically reasonable. I am doubtful about such a complicated disorder. PPN salt is known to be a rather stable cation. So my feeling would be to define a more realistic  and simple model, owing to the occurence of an inversion center on which the N atom should be located.

Anyway, an explantion of the tratment of the disorder should be included in the X-ray experimental section.

Some references concerning the diffractometer and the software used should be added. The Sheldrick's reference should be updated.

Round 2

Reviewer 1 Report

The authors have addressed the issues I had pointed out in my previous report, and they enhanced the quality of the manuscript noticeably. Some minor improvements are still necessary (see list below), but these are mostly of a straightforward kind of polishing (particularly helpful to minimize the number of queries from the editorial or typesetting office), and therefore I don´t need to see the paper again for another check.

1)      There are still various typos (e.g., line 67 “as expected” (not “es expected”); line 100 “…data are in accordance…” (“are” is missing); line 178 “…in the same plane than the metal center…” should read “…in the same plane as the metal center…”; line 365 “…-2,4,6-trifluorobenzene…” (not “…-2,4,6-trifluorobenceno…”) and some others).

2)      Line 116, there is a long line of definitions for group “R”, but the formula ([Tl(aryl)2C]) does not contain an “R”. Does “R” refer to “(aryl)” or does it refer to “C” in this formula?

3)      For the commercially available n-Butyllithium solution used, solvent and concentration should be reported in the general section.

4)      In the synthesis protocols, please double-check that all compounds used are reported with mass or volume used etc., some essential data are still missing. For example, synthesis of compound 1, line 387: “a solution of butyllithium in diethyl ether was added”, please add the concentration of this solution, the volume used (mL) and the corresponding mmol. Synthesis of compound 5, line 449: The solution of compound 3 used, which mass of compound 3 was used, which volume of solvent (dichloromethane) was used?

5)      Line 459: The change made at the fluorine atom (changed from Fp to Ft), what should that be good for?

6)      Crystallography: The ShelXL version reported now (ShelXL 92) made things even worse than before. For sure, a much younger version of ShelXL has been used. Please check the CIFs carefully, you will find the details of the refinement programs used. They may differ from one to another structure.

7)      Table 4, compound 4, temperature: 273 K. This is rather unusual. What was the purpose for collecting diffraction data slightly below room temperature? (Usually, data sets are collected at room temperature or at noticeable lower temperatures such as 200 … 100 K). Please check if 273 K is correct, and if it is correct, please add a footnote which explains this rather unusual decision for this temperature.

Author Response

Again, we thank the reviewer for his/her new revision and comments, which have also been 
considered in the last version of the manuscript.
1. There are still various typos (e.g., line 67 “as expected” (not “es expected”); line 100 
“…data are in accordance…” (“are” is missing); line 178 “…in the same plane than the 
metal center…” should read “…in the same plane as the metal center…”; line 365 “…-
2,4,6-trifluorobenzene…” (not “…-2,4,6-trifluorobenceno…”) and some others).
All the typos mentioned have been amended and the manuscript (especially the 
Experimental Section) has been carefully revised in order to amend other typos. 
2. Line 116, there is a long line of definitions for group “R”, but the formula ([Tl(aryl)2C]) 
does not contain an “R”. Does “R” refer to “(aryl)” or does it refer to “C” in this formula?
This was a mistake: the formula should be [Tl(aryl)2Cl] (with chlorine), not [Tl(aryl)2C] 
(with carbon). The expression “[Tl(aryl)2C] (R = C6Cl5 [26]…” has been replace by 
“[Tl(aryl)2Cl] (aryl = C6Cl5 [26]…”.
3. For the commercially available n-Butyllithium solution used, solvent and concentration 
should be reported in the general section.
The n-Butyllithium solution concentration (1.6 M) and solvent (hexane) have been 
included. 
4. In the synthesis protocols, please double-check that all compounds used are reported 
with mass or volume used etc., some essential data are still missing. For example, 
synthesis of compound 1, line 387: “a solution of butyllithium in diethyl ether was 
added”, please add the concentration of this solution, the volume used (mL) and the 
corresponding mmol. Synthesis of compound 5, line 449: The solution of compound 3 
used, which mass of compound 3 was used, which volume of solvent (dichloromethane) 
was used? 
All the data required have been included in the manuscript, even the mass of 3
employed in the synthesis of 4 and 5, for which only the mmol used in the synthesis had 
previously been reported since, due to the oily nature of 3, this starting complex is used 
without previous isolation, that is, from the solution in which it has been freshly 
prepared. Therefore, complex 3 is not really weighted for the synthesis of 4 and 5, and 
the mass specified in both cases has been calculated.
5. Line 459: The change made at the fluorine atom (changed from Fp to Ft), what should 
that be good for?
It was a typo that occurred during the revision process, and it has been corrected. The 
initial “Fp” was correct.
6. Crystallography: The ShelXL version reported now (ShelXL 92) made things even worse 
than before. For sure, a much younger version of ShelXL has been used. Please check 
the CIFs carefully, you will find the details of the refinement programs used. They may 
differ from one to another structure.
The ShelXL (ShelXL 2018/3) and ShelXT (ShelXT 2014/5) versions have been corrected 
according to those employed in the solution and refinement of the structures.
7. Table 4, compound 4, temperature: 273 K. This is rather unusual. What was the purpose 
for collecting diffraction data slightly below room temperature? (Usually, data sets are 
collected at room temperature or at noticeable lower temperatures such as 200 … 100 
K). Please check if 273 K is correct, and if it is correct, please add a footnote which 
explains this rather unusual decision for this temperature.
We usually collect data at low temperatures (at 100 K when possible). The higher 
temperature (273 K) at which data were collected in the particular case of complex 4
was only due to the temporary malfunction of the low-temperature system. As the final 
data were good enough as to obtain the structure with the required accuracy, we didn´t 
measure it again at a lower temperature. If the reviewer agrees with our decission, we 
do not find necessary to include a footnote in this sense. Of course, if he/her prefers we 
to include it, we will do it

Reviewer 3 Report

This revised version is greatky improved and the X-ray data are correctly presented. I understand the authors' difficulty in modelyzing th PPN  and as I couldn't improver the model, I accept the one defined by the authors

Author Response

We thank the reviewer comments and, according to his/her second revison, we will keep the proposed model for the PPN+ cation as it was in the original version of the paper.
